# lncRNA–miRNA–mRNA ceRNA Network Involved in Sheep Prolificacy: An Integrated Approach

**DOI:** 10.3390/genes13081295

**Published:** 2022-07-22

**Authors:** Masoumeh Sadeghi, Abolfazl Bahrami, Aliakbar Hasankhani, Hamed Kioumarsi, Reza Nouralizadeh, Sarah Ali Abdulkareem, Farzad Ghafouri, Herman W. Barkema

**Affiliations:** 1Environmental Health, Zahedan University of Medical Sciences, Zahedan 98, Iran; sadeghimasoome2@gmail.com; 2Department of Animal Science, College of Agriculture and Natural Resources, University of Tehran, Karaj 31, Iran; a.hasankhani74@ut.ac.ir (A.H.); farzad.ghafouri2@ut.ac.ir (F.G.); 3Department of Animal Science Research, Gilan Agricultural and Natural Resources Research and Education Center, Agricultural Research, Education and Extension Organization (AREEO), Rasht 43, Iran; hkioumarsi@yahoo.com; 4Department of Food and Drug Control, Faculty of Pharmacy, Jundishapour University of Medical Sciences, Ahvaz 63, Iran; 5Department of Computer Science, Al-Turath University College, Al Mansour, Baghdad 10011, Iraq; sarah.ali@turath.edu.iq; 6Department of Production Animal Health, Faculty of Veterinary Medicine, University of Calgary, Calgary, AB T2N 4Z6, Canada; barkema@ucalgary.ca

**Keywords:** corpus luteum, RNA-seq, lncRNA–miRNA–mRNA ceRNA network, protein–protein interaction, prolificacy, sheep

## Abstract

Understanding the molecular pattern of fertility is considered as an important step in breeding of different species, and despite the high importance of the fertility, little success has been achieved in dissecting the interactome basis of sheep fertility. However, the complex mechanisms associated with prolificacy in sheep have not been fully understood. Therefore, this study aimed to use competitive endogenous RNA (ceRNA) networks to evaluate this trait to better understand the molecular mechanisms responsible for fertility. A competitive endogenous RNA (ceRNA) network of the corpus luteum was constructed between Romanov and Baluchi sheep breeds with either good or poor genetic merit for prolificacy using whole-transcriptome analysis. First, the main list of lncRNAs, miRNAs, and mRNA related to the corpus luteum that alter with the breed were extracted, then miRNA–mRNA and lncRNA–mRNA interactions were predicted, and the ceRNA network was constructed by integrating these interactions with the other gene regulatory networks and the protein–protein interaction (PPI). A total of 264 mRNAs, 14 lncRNAs, and 34 miRNAs were identified by combining the GO and KEGG enrichment analyses. In total, 44, 7, 7, and 6 mRNAs, lncRNAs, miRNAs, and crucial modules, respectively, were disclosed through clustering for the corpus luteum ceRNA network. All these RNAs involved in biological processes, namely proteolysis, actin cytoskeleton organization, immune system process, cell adhesion, cell differentiation, and lipid metabolic process, have an overexpression pattern (Padj < 0.01). This study increases our understanding of the contribution of different breed transcriptomes to phenotypic fertility differences and constructed a ceRNA network in sheep (*Ovis aries*) to provide insights into further research on the molecular mechanism and identify new biomarkers for genetic improvement.

## 1. Introduction

Enhancing fertility will improve the efficiency of animal production and reproduction. Expansion of reproductive performance will boost the lifetime of a sheep, reduce veterinary treatments and costs, lower insemination efforts, and shorten the lambing interval [1]. Fertility is a critical factor for the sustainability of sheep farming. Although, since the 1980s, fertility in sheep and cattle has reduced and has become one of the major causes for culling and replacement [2,3,4]. Few studies have been done in the last decades to survey the genetic bases of this decline in fertility, considering the reproduction process focusing on some particular stages of reproduction. A study has shown that reproductive traits in ewe can be regulated by individual genetic markers with major effects, known as prolificacy genes [5], or by polygenic effects, specifically in prolific sheep breeds, such as Romanov [6]. Several mutations in major genes, such as *B4GALNT2, BMPR1B, BMP15*, and *GDF9,* control litter size and ovulation rates in the ewe [7,8,9]. In recent years, microarray and transcriptomic analysis through RNA-seq of different ewe reproduction tissues have furnished additional understandings of the gene expression, and a few novel genes (e.g., *IL1B, IL1A, UCP2, STAR*, and *PTGS2*) have been discovered to be associated with fecundity in the ewe [10]. Earlier studies have demonstrated that high prolificacy can result from the action of either a single gene with a major effect, as in small-tailed Han, Lacaune, Boorola Merino, and the Chinese Hu breeds [11,12,13], or various sets of genes, as in the Romanov and Finnsheep breeds [14]. The Romanov, one of the most highly prolific sheep, has been exported to more than 40 countries to improve regional sheep, although the ovulation rate heritability is low [15]. Recently, a FecGF mutation in gene *GDF9* has been recognized to be strongly associated with prolificacy in Finnsheep and other breeds [16]. The success of pregnancy in ewes and other ruminants is demarcated at the preimplantation stage, in which the corpus luteum (CL) and endometrium play vital roles [17]. Therefore, applied knowledge of molecular mechanisms was expected to help in identifying hub genes or modules and explaining interactions between fertility and other traits [18]. For example, oocyte or spermatozoa quality and their interactions, as well as embryo and fetus developmental stages, have been recognized using omics tools in different species [19]. However, the significance of RNAs in sheep fertility features remained unknown. 

Omics directs the usage and the high-throughput technologies’ output analysis to identify and interpret the genomics, transcriptomics, proteomics, metabolomics, and epigenetics levels [20]. Further, QTL (quantitative trait loci), gene mapping, and functional genomics investigations have been carried out to explore the most important genes associated with reproductive traits [21]. Unfortunately, no clear picture has appeared at omics levels for sheep prolificacy, specifically. Accordingly, reproductive success in sheep is now seen as the result of complex interactions between different factors and omics levels [22]. On the other hand, fertility has been of high economic value in recent decades, and compared with other mammals, sheep are a potential model for studying the genetic basis of reproductive traits [23,24]. Complex interactions between the fetus, ovary, corpus luteum (CL), and endometrium are affected by maintenance and facility of pregnancy. In this regard, the endometrium stimulates embryo development through secretions in the histotroph [25,26,27]. After ovulation, a critical step in the rapid ascent of progesterone (P4) concentration is the cellular reorganization of the ovulatory follicle to create a highly vascularized CL [28]. The corpus luteum (CL) is an obvious target for differentially expressed genes studies to detect hub genes between low-fertility and high-fertility breeds [29]. An integrated approach is needed to decipher the large-scale data generated with high-throughput technologies. Integrated analyses can combine multilevel views of physiology data into a holistic interpretation of nonlinear molecular procedures [30,31]. Currently, large databases of big biological/computational data are available including interactions and records of protein functions. Correspondingly, various bioinformatics tools, computational approaches, and algorithms have been designed to discover key regulatory modules in the different complex biological networks [32,33]. Regardless, few components of these interactions have been studied for decades, and the accumulation of large-scale datasets to construct networks is a unique advancement in genetics and medicine [34,35]. Moreover, the recent improvement in molecular biology data has highlighted the urgency of integrating networks such as lncRNA–miRNA–mRNA ceRNA networks. Multi-partite networks consider various RNAs and have disclosed a new mechanism of interaction between RNAs. Long non-coding RNAs (lncRNAs), a category of non-coding RNA transcripts longer than 200 nucleotides [36], are known to be involved in considerable biological procedures, such as cell proliferation and transcriptional regulation [37]. Similarly, miRNAs (micro-RNAs), a category of short non-coding RNAs, play critical regulatory roles in multiple biological procedures, such as cell differentiation, cell migration, oncogenesis, and apoptosis, by suppressing the mRNA [38]. Nonetheless, the complex mechanisms associated with prolificacy in sheep have not been fully comprehended. Interestingly, the need to use this approach to understand the molecular regulatory mechanisms behind polygenic traits and to fully dissect the etiology of complex traits becomes even more important [39,40]. Thereby, analysis of molecular pathways has the potential to illuminate the trait progression and response to treatment at the molecular level. Accordingly, the identification of hub genes, proteins, and pathways can be understood by utilizing gene/protein interaction network models. 

In this study, we conducted multi-partite networks on reproductive traits, specifically litter size in the sheep breeds of high (Romanov) and low (Baluchi) fertility, with a litter size ranging from one to four, from different geographic regions of the farms, respectively. Romanov sheep show outstanding reproduction qualities: early sexual maturity, out-of-season breeding, and extraordinary fertility. However, the Baluchi sheep is a relatively low-prolificacy breed originally from Khorasan Razavi province, Iran, and excels in fleece weights and muscle growth [41]. Therefore, a comparison of the transcriptomic level by integrative analysis and the other integrated multi-omics level in these two breeds can help us to identify related genes and ncRNAs, their functions, and the important pathways for further genetic improvement of the reproduction traits in sheep as well as other mammals.

In this way, a ceRNA network consisting of lncRNA–miRNA–mRNA was constructed based on lncRNA–mRNA and miRNA–mRNA interactions. In addition, the other networks including protein–protein interaction networks (PPI), metabolic pathways, and gene regulatory networks (GRNs) of the hub genes involved in the ceRNA network were conducted to explore the effects of functional modules and hub differentially expressed genes, miRNAs, and lncRNAs on prolificacy. Overall, our results suggest a group of genes or mRNAs, miRNAs, and lncRNAs using a computational approach that can play key and potential roles in transcriptional and post-transcriptional gene regulation under different conditions.

## 2. Materials and Methods

The complete workflow for the sample collection, preparation, sequencing, and analysis of relevant species related to the corpus luteum in two sheep breeds is presented in Figure 1.

### 2.1. Sample Collection 

A total of 48 ewes from two sheep breeds of high prolificacy (Romanov, *n* = 24) and one of low prolificacy (Baluchi, *n* = 24) were collected from University of Tehran scientific park in Iran. The sheep included were as unrelated as possible based on records for the phenotype of litter size and the other reproductive traits. Data for the phenotype of litter size records are presented in Appendix A. 

### 2.2. Ovulation Synchronization

Sheep were synchronized based on a protocol (CIDR) described by Herlihy et al. [42] and CL samples were collected at appointed dates. On day −10, each sheep was administered an i.m. injection of a gonadotropin-releasing hormone (GnRH) agonist, and an internal drug release device (CIDR; Pfizer; Germany) was inserted per vaginum. Then, on day −3, each sheep was administered an i.m. injection of prostaglandin F2α (PGF2α) (Lutalyse; Pfizer; Germany). Then, on day −2, the CIDR device was again removed, and 24 h later, each sheep was administered a second i.m. injection of GnRH agonist.

### 2.3. Tissue Biopsies

On day 10 of the estrous cycle, CL biopsies were collected from each sheep as described previously [43,44]. Briefly, sheep were sedated with intravenous xylazine, and caudal epidural anesthesia was induced to prevent abdominal straining. The luteal biopsy was performed using a tissue biopsy needle (Ovum Pick-up instrument equipped with a 48 cm long trocar tip; SABD-1648-15-T; US Biopsy; Canada). In the biopsy sample, the mean luteal tissue area was 425 mm^2^ (range, 313 to 524 mm^2^). Some corpora lutea contained a central luteal cavity with an area of 335 mm^2^. A total of 59 biopsy attempts were made; 54 (92%) resulted in obtaining a luteal tissue specimen at the first attempt. The average size of the obtained biopsy core was 1 mm in diameter and 10.8 mm in length (range, 10 to 12 mm), and the average weight was 4.4 mg (range, 1.8 to 7.5 mg). CL samples were instantly frozen in liquid nitrogen and stored at −80 °C.

### 2.4. RNA Extraction

Whole RNA was extracted from CL (HF and LF) samples using a Trizol-based method [45]. Whole RNA was purified using the miRNeasy kit (Qiagen; Hilden, Germany) for removing any DNA contamination. The RNA quality and concentration were determined using a NanoDrop ND-1000 spectrophotometer (NanoDrop Technologies LLC; Wilmington, DE, USA). The 260/280 nm ratio of absorbance ranged from 1.85 to 2.13 for all CL samples.

### 2.5. cDNA Library Preparation and Sequencing

The RNA samples were converted to cDNA libraries for sequencing based on the protocol of the Illumina TruSeq RNA Sample Preparation Kit (Illumina; San Diego, CA, USA). RNA-Seq libraries were amplified by 11 cycles of PCR. Library concentration and quality were determined by Qubit (Invitrogen; Corston, UK) and Bioanalyser 2100 (Agilent Technologies; Santa Clara, CA, USA). Each sample was sequenced on a single lane over a total of two flow cells on the Illumina HiSeq 2500 platform to generate 60 million 50-base paired-end reads, and FASTQ files were created using CASAVA (v1.9) (Illumina).

### 2.6. Quality Assessment and Adapter Trimming of Raw Sequencing Data

A quality check of the raw sequence data was accomplished using FastQC software (v0.11.9) [46]. For this purpose, the raw RNA-seq reads were imputed into the software and the sequence quality, basic statistics, sequence content, and quality score were evaluated per base. The raw sequence data that had the needs of quality features were utilized for the next analysis. Next, PCR primers, the adapters, and non-informative sequences were trimmed using the Trimmomatic software (v0.38.1) [47].

### 2.7. Sequence Alignment and Detection of RNAs

Alignment sequences, mapping, and identification of known and novel RNAs of reads were conducted on the Ovis aries reference genome (http://ftp.ensembl.org/pub/release-103/fasta/ovis_aries/dna/ (accessed on 29 December 2020)) using HISAT2 software (v2.2.1) [48]. 

### 2.8. Analyzing Differentially Expressed RNAs

For transcript quantification, featureCounts software (v2.0.1) was utilized to calculate the total raw counts of mapped reads [49]. Next, to examine whether the accumulation or degradation of transcripts was related to prolificacy, the transcripts and their expression levels were compared between corpus luteum samples of Romanov and Baluchi sheep. Differentially expressed transcripts (DETs) were performed from reading counts using DESeq2 software (v2.11.40.7) [50]. For this step, normalization of the data was performed in such a way that the raw read count of each transcript was multiplied by the sample size factor, which was calculated as a ratio of the observed count. For each transcript in each corpus luteum sample, the observed count is the ratio of the raw count for each transcript to the geometric mean across the samples. At the end of the analysis, transcripts with a false discovery rate (FDR) <0.05 and log2 fold change difference data were performed, and those with inadj <0.01 were considered as differentially expressed mRNAs and lncRNAs. For miRNA identification, quality control was performed using FastQC with default parameters. Clean tags were obtained by removing low-quality reads. Meanwhile, the clean tags were mapped to the reference genome using Bowtie2 [51]. The mapping percentage ranged from 66.45% to 80.02%, with an average of 76.74%. Differentially expressed miRNAs were identified using the generalized linear model implemented in edgeR [52] software, with thresholds FDR < 0.05 and fold-change difference ranging from 66.45% to 80.02%, with an average of 76. For the Seq results, an IGV 2.3 (Integrative Genomics Viewer) tool was used [53].

### 2.9. Validation of RNA-Seq Results Using Quantitative Real-Time PCR

To validate the reproducibility of RNA-seq data, four DEGs including *BMP2*, *HNF4A*, *PLCB2*, and *RPS6KL1* were selected for analysis by qRT-PCR. The same RNAs extracted from the CL tissues at each breed were used for qRT-PCR validation. The primer pairs were designed using Geneious Prime software v2021.1 (Appendix A). Quantitative reverse-transcription PCR was carried out according to the manufacturer’s specifications for reference to SYBR^®^ Premix Ex TaqTM. SYBR Green PCR cycling was denatured using a program of 95 °C for 10 s, and 35 cycles of 95 °C for 5 s, and 60 °C for 40 s, and performed on an ABI 7500 instrument (USA). The specificity of each PCR product was confirmed by melting curve analysis. All qRT-PCR assays were performed in triplicate reactions. The housekeeping genes RPL19 and GAPDH (glyceraldehyde-3-phosphate dehydrogenase) were used as the internal control genes. The expression levels of target mRNAs were obtained based on RNAs extracted from the four ewes and were shown to be normalized to GAPDH.

### 2.10. Gene Annotation

Gene set annotation enrichment analysis was also carried out using the online programs g: Profiler [54] (https://biit.cs.ut.ee/gprofiler/gost (accessed on 12 March 2022)), GeneCards (www.genecards.org/ (accessed on 12 March 2022)), the STRING database [55] (https://string-db.org (accessed on 12 March 2022)), and the DAVID [56] (Database for An-notation, Visualization, and Integrated Discovery; https://david.ncifcrf.gov/ (accessed on 12 March 2022)), which provide a set of functional annotation tools for the genes categorized using gene ontology (GO) terms.

### 2.11. Target Prediction and Validation of Differentially Expressed mRNAs, lncRNAs, and miRNAs

The predicted and validated mRNA genes were identified using miRWalk 3.0 (http://129.206.7.150/ (accessed on 23 December 2021)), a comprehensive atlas of microRNA–target interaction tools, which integrates 12 miRNA–target prediction tools. Further, lncRNA–mRNA interactions were predicted by the NONCODE database [57] (http://www.noncode.org/ (accessed on 3 January 2021)) and the LNCipedia database [58] (https://lncipedia.org (accessed on 3 January 2021)).

### 2.12. lncRNA–miRNA–mRNA ceRNA Network Construction and Gene Ontology

lncRNA–miRNA–mRNA ceRNA network was constructed based on miRNA–mRNA and lncRNA–mRNA interactions and online interaction databases. The Pathway Resource List (http://pathguide.org (accessed on 15 March 2022)) is a meta-database that provides more than 300 web-accessible network databases and biological pathways [59]. PPI (protein–protein interaction) data were abstracted from the Database of Interacting Proteins [60], BIND (Biomolecular Interaction Network Database) [60], MIPS (Mammalian Protein-Protein Interactions Database) [61], and BioGRID (Biological General Repository for Interaction Datasets) [62]. In addition, interaction data were obtained from the analysis of related experiments and search in interaction databases such as Gene-MANIA [63] (https://genemania.org/ (accessed on 15 March 2022)) and the STRING database [64] (Search Tool for the Retrieval of Interacting Genes or Proteins; https://string-db.org (accessed on 15 March 2022)). To describe the interaction between proteins using a probabilistic confidence score, the string uses eight major sources of interaction or association data, including co-occurrence, neighborhood, fusion, co-expression, experimental, text mining, and database [64].

Moreover, several Cytoscape plugins were used for different purposes such as screening, integrating, visualizing, and analyzing interactive data. In the respective networks, molecular species (RNAs) and the interactions between them are represented as nodes and edges, respectively. Gene Ontology terms with FDR < 0.05 were considered significantly enriched for the identified genes. Furthermore, constructed networks were compiled in simple interaction format (SIF) bowed to Cytoscape for topological analysis. Then, the statistical and topological significance of the network was assessed using the Network Analyzer plugin in Cytoscape software (v3.8.2.) [1] (National Institute of General Medical Sciences, Bethesda Softworks, Rockville, MD, USA). The mean path length, the degree of nodes, the network diameter, and the shortest path lengths between any two nodes compared with random networks (Barabasi-Albert and Erdos-Renyi models) were analyzed.

### 2.13. Clustering of lncRNA–miRNA–mRNA ceRNA Network

Topological properties of the lncRNA–miRNA–mRNA ceRNA network were evaluated by Cytoscape software (v3.8.2.) and, for clustering, ClusterONE [65] and MCODE [66] were used. ClusterONE was developed to discover densely connected sub-graphs of a network by minimizing edges between different clusters and maximizing edges within a cluster. MCODE is a clustering algorithm, which can be used for directed or undirected graphs.

## 3. Results

### 3.1. Candidate lncRNA, miRNA, and mRNA/Gene List

A summary of the RNA-seq data analysis pipeline and the steps of constructing of lncRNA–miRNA–mRNA ceRNA network is shown in Figure 1. A total of 264 genes/mRNAs, 14 lncRNAs, and 34 miRNAs in the corpus luteum samples of Romanov compared with Baluchi sheep were identified from RNA-Seq data. The total RNAs in these two breeds that are known to be involved in prolificacy are given in Appendix A. These RNAs are annotated and described based on the molecular and biological processes in the Gene Ontology (GO) databases. For validation of alignment related to transcripts, the result of aligned reads corresponding to one of the key genes of interest (AGR2) is shown in Appendix A.

### 3.2. Global Transcriptome Was Differentially Expressed in Romanov Compared with Baluchi Sheep

To discover the prolificacy of the mRNA–lncRNA–miRNA regulatory network, we first analyzed the global transcriptome in CL in both Romanov and Baluchi sheep breeds. Using the hierarchical clustering algorithm, we detected that the global gene expression is changed in CL (Appendix A), with 155 and 3 mRNAs/genes and lncRNAs significantly up-regulated (FDR < 0.05, fold-change > 1.5) and 109 and 11 mRNAs/genes and lncRNAs significantly down-regulated (FDR < 0.05, fold-change < −1.5), respectively. We identified a total of 34 miRNAs altogether. Moreover, we identified 19 up-regulated and 15 down-regulated miRNAs by comparing the gene expression profile of Romanov with Baluchi through applying stringent filtering criteria (false discover rate, FDR < 0.05, fold-change > 1.5, and *p* < 0.01) (Appendix A). 

### 3.3. Analysis of Expression based on RT-qPCR Data

To assess the accuracy and reliability of differentially expressed genes identified by RNA-seq, four DEGs from two breeds were selected to perform qRT-PCR tests. The expression results for four genes assessed using RNA-seq and qRT-PCR are shown in Figure 2. As can be observed, the expression patterns of four genes showed a general agreement between the two technologies. 

### 3.4. GO and Pathway Analysis of Differentially Expressed mRNAs

GO analysis was used to explore the function of the significantly altered mRNAs/genes. The differentially expressed mRNAs were mainly enriched in six molecular function terms containing proteolysis, actin cytoskeleton organization, immune system process, biological adhesion, cell differentiation, and the lipid metabolic process. The KEGG, WikiPathways, and Reactome analysis was performed to identify pathways that were significantly enriched with differentially expressed mRNAs (*p* < 0.05) (Figure 3A). Among these pathways, the Wnt signaling pathway contained the largest number of differentially expressed genes.

### 3.5. Construction of lncRNA–miRNA–mRNA ceRNA Network

To discover the mechanism of how lncRNAs regulate mRNA through sponging miRNA, a ceRNA network was constructed with a combination of predicted miRNA–mRNA and lncRNA–mRNA interactions. Based on the knowledge of interactions in databases, we could detect interactions for 312 nodes and 748 edges in the network. Moreover, the related files of networks are given in Appendix A. Finally, 264 differentially expressed mRNAs, 14 differentially expressed lncRNAs, and 34 differentially expressed miRNAs were included in the network. To further explore the most significant clusters of mRNAs/genes in the ceRNA network, we conducted PPI network, BIND, MIPS, and BioGRID. In addition, interaction data were extracted from the analysis of related experiments and search in interaction databases such as GeneMANIA and the STRING database. As mentioned, molecular species (mRNAs, lncRNAs, and miRNAs) in constructed networks are indicated as nodes and the interactions between them as edges. Moreover, constructed networks were compiled in simple interaction format (SIF) bowed to Cytoscape (v3.8.2.) (National Institute of General Medical Sciences, Bethesda Softworks, Rockville, MD, USA) for topological analysis. 

### 3.6. Topology Analysis

Network characteristics were calculated using the Network Analyzer plugin of Cytoscape for the ceRNA Network. Topological analysis recognizes the qualitative virtues of the complex biological systems. Network topology is used to examine the state of communication and information transfer of a node with other nodes of interactive networks. Topological parameters such as the average clustering coefficient of degrees, topological coefficient, average degree, betweenness centrality, and power–law distribution were evaluated. The distribution of the clustering coefficient is an important parcel of biological scale-free networks. Then, network density, characteristic path length, network centralization, and the clustering coefficient of the ceRNA network were compared with randomized model networks (Barabasi–Albert and Erdos–Renyi models), as presented in Table 1. The structure of the ceRNA network is far from the structure of the simulated randomized network. In particular, the clustering coefficient of the ceRNA network is significantly different from the random network.

### 3.7. Clustering of lncRNA–miRNA–mRNA ceRNA Network

Finally, we utilized clustering on the lncRNA–miRNA–mRNA ceRNA network made by an integrated approach. Clustering algorithms are utilized to determine significant sub-networks or modules. The results were clustered by ClusterONE [65] and MCODE software [66]. ClusterONE output was 19 modules or clusters including 68 nodes. MCODE output was 13 modules including 89 nodes (considering overlapping cluster nodes). Some of the clusters turned out to be sub-clusters of other larger clusters, so such sub-clusters were removed and the final number of clusters was diminished from 13 to 6 modules. Many of the nodes are repeated in more than one cluster and, in total, there were 44, 7, and 7 unique mRNAs/genes, lncRNAs, and miRNAs, respectively, out of six modules (Figure 4, Figure 5, Figure 6, Figure 7, Figure 8 and Figure 9). These clusters are shown in Appendix A. For constructing a pathway network, different databases were integrated and percentages of visible nodes at each term are shown in Figure 4. 

## 4. Discussion

The whole-transcriptome profiling technique helps gain a more in-depth understanding of the functions of the corpus luteum, which may authorize the marker identification that is differentially expressed, for instance, between two breeds showing different litter size phenotypes. Although most of the investigations associated with pregnancy have been performed in sheep, only a few studies have used the whole-transcriptome approaches to the corpus luteum. Microarray-based transcriptomic research performed by Gray et al. (2006) identified several endometrial genes regulated by progesterone (from the corpus luteum). In a more comprehensive investigation performed by Brooks et al. [67], transcriptome analysis of corpus luteum during the peri-implantation period of pregnancy recognized different biological functions and regulatory pathways in sheep. Moore et al. [68] integrated GWAS with gene expression data to understand the roles of the corpus luteum transcriptomes in cattle fertility. Kfir et al. [69] recognized DEGs (differentially expressed genes) between day 4 and day 11 in the CL in dairy cows. Furthermore, an investigation of endometrial transcriptome differences between European mouflon and Finnsheep identified several transcripts associated with fertility [70]. Most studies just identify DEGs without recognizing associations among mRNAs/genes and considering other biological molecules including lncRNAs and miRNAs, whereas considering the interactions among all regulatory factors can provide an exhaustive understanding of the mechanisms involved in reproductive traits. 

In 2011, Salmena et al. suggested the ceRNA (competitive endogenous RNA) hypothesis that lncRNAs and protein-coding RNAs can operate as competitive endogenous RNAs to communicate by binding to miRNA sites [71,72]. According to the competitive endogenous RNAs hypothesis, many researchers have dedicated themselves to elucidating the competitive endogenous RNA roles of lncRNAs in some traits by constructing competitive endogenous RNA networks [73,74]. However, the entire regulatory network that links the processes of non-coding RNAs and coding has not been extensively explained. Collecting evidence proposes that lncRNA may operate as a competitive endogenous RNA for certain miRNAs to modulate the target mRNAs/genes of the miRNAs [75]. Several studies aiming to construct lncRNA-associated competitive endogenous RNA networks in production traits have been performed [76,77,78]. In this study, we first identified 264 differential expressed mRNAs, 34 differential expressed miRNAs, and 14 differentially expressed lncRNAs between CL samples of two different breeds. We finally constructed an lncRNA–miRNA–mRNA ceRNA network including 312 nodes and 748 lncRNA–miRNA–mRNA ceRNA interactions. Although we recognized potential mRNA–lncRNA and miRNA–mRNA interactions involved in prolificacy development by constructing a multi-partite network, a limitation in our analysis should be mentioned. The potential interactions were identified by prediction tools and RNA-seq; therefore, in the future, more experimental studies should be performed to validate the target predictions in sheep prolificacy development. The differentially expressed RNAs in the corpus luteum were involved in processes associated with proteolysis, actin cytoskeleton organization, immune system process, biological adhesion, cell differentiation, and lipid metabolic process. The computational method suggested in this study goes further by associating the various phases with ceRNA regulatory modules. The lncRNA–miRNA–mRNA ceRNA network defined in this study consists of mRNAs/genes, miRNA, and lncRNA and associated conditions in the indicated case. This potential approach provides a promising new opportunity to evaluate genetic traits such as the reproductive process by discovering ceRNA regulatory networks by linking different types of RNAs to different breeds or different groups. Understanding the molecular mechanism of fertility is needed to solve the complex synergies orchestrated during the process of reproduction. Although, several studies have described the presence of altered gene expression patterns in different reproductive processes [79,80,81,82]. 

Herein, we discovered several mRNAs/genes in the corpus luteum of high- and low-fertility sheep. Indeed, the role of these genes during fertility is not clearly understand yet, but some mRNAs/genes family members are believed to be involved in cell survival, proliferation, apoptosis, and other fundamental biological processes. The expression of these mRNAs/genes in both high-fertility and low-fertility in the corpus luteum could show their basal function cellular roles during the reproductive performance [83]. Our results have detected six candidate modules involved in sheep fertility, which are available in detail in Appendix A. These modules have detected 44, 7, and 7 mRNAs/genes, lncRNAs, and miRNAs (without repeated nodes), respectively, and modules 1, 2, 3, 4, 5, and 6 included 41, 20, 17, 78, 37, and 19 nodes, respectively, and, 129, 58, 42, 174, 60, and 22 interactions, respectively. Signaling pathway, biological process, and GO enrichment analysis of deregulated RNAs specified important biological processes, which are dysregulated by deregulated RNAs.

In module 1, let-7b, let-7a, let-7c, and miR-199a suppressed NOXA1, SYNGAP1, ROCK2, and PLCB2 genes, and almost all miRNAs were up-regulated, except let-7c, which was down-regulated and genes were down-regulated. Moreover, in this module, ENST00000416770, ENST00000567819, ENST00000559520, and ENST00000618857 interacted with RPS6KA6, CAPN3, PLCB2, and BACE1, respectively. The hub-hub genes included CAPN3, PLS1, CNN1, ACTA1, and KLHL17, in which PLS1, CNN1, and ACTA1 genes were up-regulated and CAPN3 and KLHL17 genes were down-regulated. All of these hub-hub genes are involved in the actin cytoskeleton organization process. Calpain 3 (CAPN3), a muscle-specific member of the calpain large subunit family that specifically binds to titin, has been suggested to be related to muscle growth in cattle and broiler chickens. Gene Ontology (GO) annotations related to this gene include calcium ion binding, peptidase activity, meat traits, and disease resistance [84]. PLCB2 gene interacted with ENST00000559520 and both of them were down-regulated. Moreover, this gene was suppressed by miR-199a. PLCB2 encodes the protein that belong to phospholipases group and hydrolyze phospholipids into fatty acids and other lipophilic molecules. In animals, plastin 1 (PLS1) is a component of membrane signaling complexes controlling cell differentiation, motility, and adhesion, and has effects on calcium ion and actin filament binding [85]. Calponin 1 (CNN1) contributes to regulating contractility. This gene encoded cytoskeletal signaling and endothelin pathways. Moreover, it affects actin-binding and calmodulin-binding and is specifically expressed in smooth muscle cells [86,87]. Actin α 1 (ACTA1) gene encodes the thin filaments of the muscle contractile apparatus. This gene has a role in the structural constituent of the cytoskeleton, myosin binding, RhoGDI pathway, and cytoskeleton remodeling regulation of actin cytoskeleton by Rho GTPases [88]. Kelch like family member 17 (KLHL17) is an actin-binding protein that has a role in actin filament binding and POZ domain binding and is predominantly expressed in neurons of most regions of the brain [89]. A process that is fulfilling at the cellular level results in the arrangement, assembly of constituent parts, or disassembly of cytoskeletal structures comprising actin filaments and their related proteins [90]. 

In module 2, *C3, SERPINE1, CFB, C5*, and *CFI* genes were identified as the most important genes in that all of them were upregulated. The key biological process affected by these genes was the proteolysis process. The hydrolysis of proteins into amino acids or/and smaller polypeptides occurred by cleavage of their peptide bonds [91]. Serpin Family E Member 1 (SERPINE1) is a protein coding gene that has been determined to be transiently upregulated during the latter part of the estrous cycle and early luteolysis. Moreover, it may have a role in signaling receptor binding, protease binding, regulation of the extracellular matrix to facilitate the invasion of immune cells, and inhibiting the synthesis of progesterone [92]. Complement components C3 and C5 play a central role in the activation of the complement system. C5 (complement C5) is a gene encoding a protein of the innate immune system that had an important role in inflammation, host homeostasis, and host defense against pathogens. Some results present that many factors in the complement system (including C1, C3, C5, C6, C7, C8, and C9, as well as complement factor B and factor H) are regulated by heat stress in the blood [93]. Moreover, the complement system is a part of the body’s immune system, which plays a key role in immune responses and defense against infection [94]. Complement factor B (CFB) of the alternative complement pathway has been identified in sheep as a major gene that has a key role in the regulation of the immune reaction, serine-type endopeptidase activity, and complement activation [95,96]. Complement Factor I (CFI) is a protein-coding gene that is related to the following pathways: immune response lectin induced complement pathway and the innate immune system [97].

In module 3, let-7c suppressed the C3 gene and was down-regulated, and the mentioned gene was up-regulated. The hub-hub key genes involved in the immune system process were *C4BPB, SERPINE1, CFB, C4BPA,* and *MASP2. C3, C4BPB, SERPINE1, CFB*, and *C4BPA* genes were upregulated and *MASP2* gene was downregulated. *C4BP* gene binds strongly to apoptotic and necrotic cells and limits their complement activation. Complement component 4 binding protein β (C4BPB) and complement component 4 binding protein α (*C4BPA*) genes encode a member of a superfamily of proteins composed predominantly of tandemly arrayed short consensus repeats of approximately 60 amino acids. This gene has a major role in the innate immune system and immune response lectin induced complement pathways [98]. MBL associated serine protease 2 (*MASP2*) gene is associated with biological processes terms including acute inflammation and defense response. Activities related to this gene include calcium ion binding and peptidase activity [99]. Therefore, the biological process of these genes is involved in the functioning or development of the immune system and organismal system for calibrated responses to invasive or internal threats [100].

In module 4, miR-21, let-7b, let-7a, miR-16b, let-7c, let-7f, and miR-199a were identified. miR-21 suppressed *CDK3, KLHL17, MAPK15*, and *OLR1* genes and was downregulated. let-7b suppressed *CDK3, GNB3, HOMER1, KLHL17, OLR1*, and *PRKACA* genes and was upregulated. let-7a suppressed *ERBB3* and *NME2* genes and was up-regulated. miR-16b suppressed the *DAPP1* gene and was up-regulated. let-7c suppressed the *MYB* gene and was down-regulated, and the *MYB* gene was up-regulated. let-7f suppressed *CAPN3, CDH1*, and *SPEG* genes and was down-regulated. miR-199a suppressed *ALS2CL, MAPK15*, and *SERPINE1* genes and was up-regulated. Moreover, in this module, ENST00000559520, ENST00000586450, ENST00000501818, ENST00000455642, and ENST00000567819 interacted with *PLCB2, MAP3K14, FABP7, HNF4A*, and *CAPN3*, respectively. The hub-hub genes included *ITGA2B, CDH1, SERPINE1, BMP2, TNC, NRCAM, CLDN3, CLDN4, MYB*, and *ERBB3*. Among these, *CDH1, SERPINE1, BMP2, TNC, NRCAM, CLDN3, CLDN4, MYB*, and *ERBB3* genes were upregulated and the *ITGA2B* gene was downregulated. Integrin subunit α 2b (ITGA2B) is a protein-coding gene that has a role in cytokine signaling in the immune system and RET signaling pathways and activities related to this gene including identical protein binding and fibrinogen binding. Moreover, for adhesion with this gene, integrins are known to participate in cell-surface mediated signaling [101]. Estradiol secretion by granulosa cells is generally increased in the presence of bone morphogenetic protein 2 (BMP2) in mammals [102,103]. Tenascin C (TNC) gene encodes an extracellular matrix protein with a spatially and temporally restricted tissue distribution. This gene has been reported to be involved in vertebrate neural, skeletal, and vascular morphogenesis during development as well as in the regulation of neuronal differentiation in the nervous system [104,105]. Neuronal cell adhesion molecule (NRCAM) is an Ig superfamily adhesion molecule that has a major role in L1CAM interactions and developmental biology pathways and ankyrin binding and protein binding are involved in heterotypic cell–cell adhesion [106]. Claudin 3 (CLDN3) is a protein-coding gene that has a key role in the blood–brain barrier and immune cell transmigration: VCAM-1/CD106 signaling and cell junction organization. Moreover, CLDN3, CLDN4, and MYB genes are involved in modules associated with reproductive and fertility complex traits [105]. Erb-B2 receptor tyrosine kinase 3 (ERBB3) is a member of the epidermal growth factor receptor (EGFR) family of receptor tyrosine kinases that have a role during fetal ovary development in cows. Among its related pathways are NF-kappaB pathway and cytokine signaling in the immune system [107]. All of these genes are involved in the biological adhesion process. This involves the attachment of an organism/cell to a substrate/other organisms. Biological adhesion includes intracellular attachment between membrane regions. 

In module 5, miR-21, let-7b, miR-16b, let-7c, let-7f, and miR-199a were identified. miR-21 suppressed *BMP2, MAP3K14, MAP3K7*, and *RPS6KA6* genes and was down-regulated. let-7b suppressed *MAP3K14* and *TRAF6* genes and was up-regulated. miR-16b suppressed the *MAP3K7* gene and was up-regulated. let-7c suppressed the *HSF4* gene and was down-regulated. let-7f suppressed the *ATF3* gene and was down-regulated. miR-199a suppressed *BMP2* and *RPS6KA6* genes and was up-regulated. Moreover, in this module, ENST00000455642, ENST00000416770, and ENST00000586450 interacted with *HNF4A, RPS6KA6*, and *MAP3K7*, respectively. *ATF3, HSF4, COL27A1, MYB, MAPT, SPIB, SFRP2, GREM1*, and *BMP2* genes were identified as hub-hub genes. Among these, *ATF3, MYB, SPIB, SFRP2*, and *GREM1* genes were up-regulated and *HSF4*, *COL27A1*, and *MAPT* genes were down-regulated. The key biological process affected by these genes was cell differentiation. The process includes unspecialized cells, for example, functional features that characterize the cells, embryonic/regenerative cells, and tissues/organs of the mature organism. Differentiation includes the processes involved in the obligation of a cell to a specific fate and its further development to the mature state [108]. Activating transcription factor 3 (ATF3) is an estrogen-responsive gene, a member of the mammalian activation transcription factor/cAMP-responsive element-binding (CREB) protein family of transcription factors. This gene has a role in PERK regulating gene expression and unfolded protein response (UPR) pathways [109]. Collagen type XXVII α 1 chain (COL27A1) gene is a member of the fibrillar collagen family and has a role in the calcification of cartilage and the transition of cartilage to bone [110]. Microtubule associated protein tau (MAPT) gene promotes microtubule assembly and stability and might be involved in the establishment and maintenance of neuronal polarity [111]. Gremlin 1, DAN family BMP antagonist (GREM1) gene, encodes a member of the BMP (bone morphogenic protein) antagonist family and has a role in cytokine activity, BMP binding, and reproductive traits in Awassi sheep [112]. In this module, *MAP3K14* and *RPS6KA6* interacted with ENST00000586450 and ENST00000416770, respectively. All of them were down-regulated and were suppressed by miR-16b and miR-199a, which were up-regulated. *MAP3K14* encodes mitogen-activated protein kinase kinase kinase 14, which is a serine/threonine protein-kinase. This kinase binds to TRAF2 and stimulates NF-kappaB activity. It shares sequence similarity with several other MAPKK kinases. It participates in an NF-kappaB-inducing signalling cascade common to receptors of the tumour-necrosis/nerve-growth factor (TNF/NGF) family and to the interleukin-1 type-I receptor. Regardless, *RPS6KA6* encodes protein that constitutively activate serine/threonine-protein kinase, which exhibits growth-factor-independent kinase activity and may participate in p53/TP53-dependent cell growth arrest signaling and play an inhibitory role during embryogenesis.

In module 6, let-7f suppressed *FABP7* and *HNF4A* genes and was downregulated. Moreover, in this module, ENST00000455642 and ENST00000501818 interacted with *HNF4A* and *FABP7*, respectively. Hub-hub genes were *LPIN1, HSD17B1, PLA2G4B, CYP2S1, HNF4A, TTR,* and *CRABP2*. Among these, *HSD17B1, CYP2S1, HNF4A,* and *CRABP2* genes were upregulated and *LPIN1* and *PLA2G4B* genes were downregulated. Lipin 1 (LPIN1) is a member of the lipin family of proteins that has an important role in animal and poultry lipid metabolism and its regulation. This gene may have value as a genetic marker for improving some traits such as meat production and carcass [113]. The *HNF4A* gene interacted with ENST00000455642 and was highly upregulated, and suppressed by let-7f. This gene is a transcriptional regulator that controls the expression of some genes during the transition of endodermal cells to progenitor cells, facilitating the recruitment of RNA pol II to the promoters of target genes and activating the transcription of ***CYP2C38***. It represses the CLOCK-ARNTL/BMAL1 transcriptional activity and is essential for circadian rhythm maintenance and period regulation in the some tissues cells. Hydroxysteroid 17-β dehydrogenase 1 (*HSD17B1*) is a gene that has a key role in the super pathway of steroid hormone biosynthesis and metabolism pathway. Moreover, the HSD17B1 gene plays an essential role in the biological processes related to sheep ovulation [114]. Phospholipase A2 group IVB (*PLA2G4B*) gene encodes a member of the cytosolic phospholipase A2 protein family and has a role in calcium ion binding and phospholipase activity. This gene has been identified as one of the important genes in interaction networks in sheep fertility [115]. Cytochrome P450 family 2 subfamily S member 1 (*CYP2S1*) encodes a member of the cytochrome P450 superfamily of enzymes. *CYP2S1* gene is related to the reproductive processes as a regulatory factor in follicular development and ovarian development in the ovulation process in female mammals [116]. Therefore, most of these genes are involved in the lipid metabolic process. The chemical pathways and reactions involving lipid compounds soluble in an organic solvent include neutral fats, fatty acids and other fatty-acid esters, sphingoids and other long-chain bases, long-chain alcohols, waxes, phospholipids, glycolipids, and the other isoprenoids [117]. 

We offered a computational approach to an lncRNA–miRNA–mRNA ceRNA network using predicted and validated expression profiles of RNAs. Indeed, the use of the same breed might lead to more accurate results. However, these two sheep breeds are different, but in systemic studies, thresholds are usually considered that have the most genetic and phenotypic differences. The spatio-temporal differential expression in different tissues, especially CL, supports the potential role of RNAs in the transcriptional and post-transcriptional regulation of genes involved in prolificacy. The reports we presented here for the first time may be beneficial in discovering the basic molecular regulatory mechanisms of fertility in sheep. Although, further efforts are needed to discover the specific biological functional role of RNA modules during various stages of the reproductive traits. In addition, our findings showed that the integration of mRNAs, miRNAs, and lncRNAs based on the ceRNA networks, along with different data such as lncRNA interactions, miRNA target prediction, PPI, and other regulatory networks, can be considered a robust approach to provide greater insights into biological processes at the molecular level.

## 5. Conclusions

This study applied a new approach to integrating different classes of RNA as an integrated network, and first reported transcriptome sequencing analysis of the corpus luteum in sheep. Integration of transcriptomic data for obtaining and identifying hub nodes with differences in expression levels led to the successful identification of 264 mRNAs/genes, 14 lncRNAs, and 34 miRNAs in the main process of prolificacy in the corpus luteum samples of Romanov compared with Baluchi breeds. Using lncRNA–miRNA–mRNA ceRNA network analysis and constructing the networks by merging the interactions and regulation networks, 6 key modules, 44 mRNAs/genes, 7 lncRNAs, and 7 miRNAs were identified as being involved in major biological processes including proteolysis, actin cytoskeleton organization, immune system process, biological adhesion, cell differentiation, and lipid metabolic process, and have an overexpression pattern. Overall, our results in this research demonstrate that integrated network analysis and the application of omics data of sheep corpus luteum generate novel insights into sheep fertility. Furthermore, a comparison of the transcriptomic level in these two breeds will be important for further genetic improvement of the trait and a better understanding of the molecular basis of reproduction in sheep as well as other mammals.

## Figures and Tables

**Figure 1 genes-13-01295-f001:**
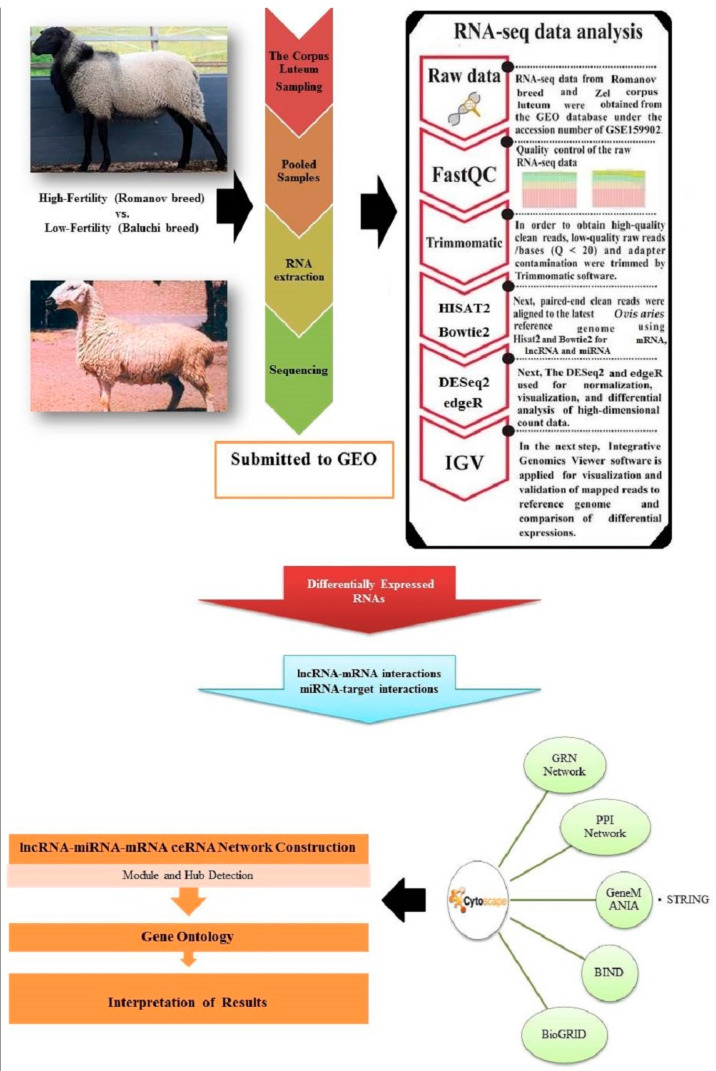
Schematic of the pipeline used to construct lncRNA–miRNA–mRNA ceRNA network of the corpus luteum in sheep. The main RNAs were identified from the RNA-Seq dataset between two different breeds. The protein–protein interaction network (PPI) and gene regulatory network (GRN) were constructed and visualized using Cytoscape 3.9.1.

**Figure 2 genes-13-01295-f002:**
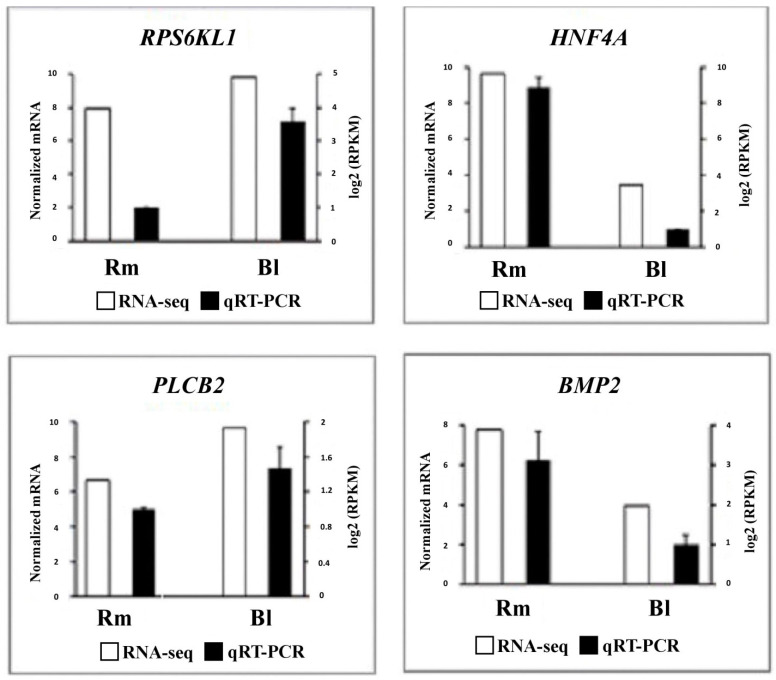
Differentially expressed genes were selected from transcriptome comparison combinations at different breeds. The black-filled columns represent the relative mRNA expression levels obtained by qRT-PCR, which were normalized by GAPDH; the blank columns show the log10 (RPKM) value obtained by RNA-seq. Rm and Bl represent Romanov and Baluchi breeds, respectively.

**Figure 3 genes-13-01295-f003:**
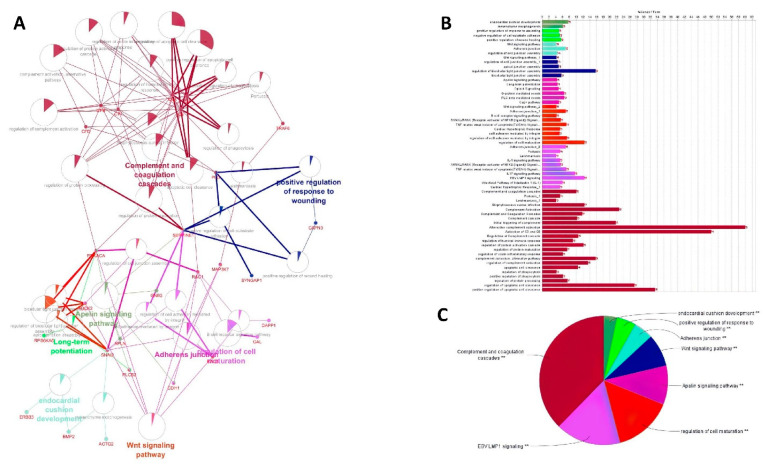
(**A**) Gene Ontology was constructed by integrating KEGG, Reactions, and WikiPathways. Each node illustrates a pathway or term and the percentage of visible shared genes between pathways or terms and edges present relationships between pathways; (**B**) bar chart shows the number of genes and related pathways using ClueGO and CluePrdia software based on different databases (**C**) pie chart shows the number of genes and related pathways using ClueGO and CluePrdia software based on different databases. ** represents hub genes were significantly enriched in these biological processes and molecular functions.

**Figure 4 genes-13-01295-f004:**
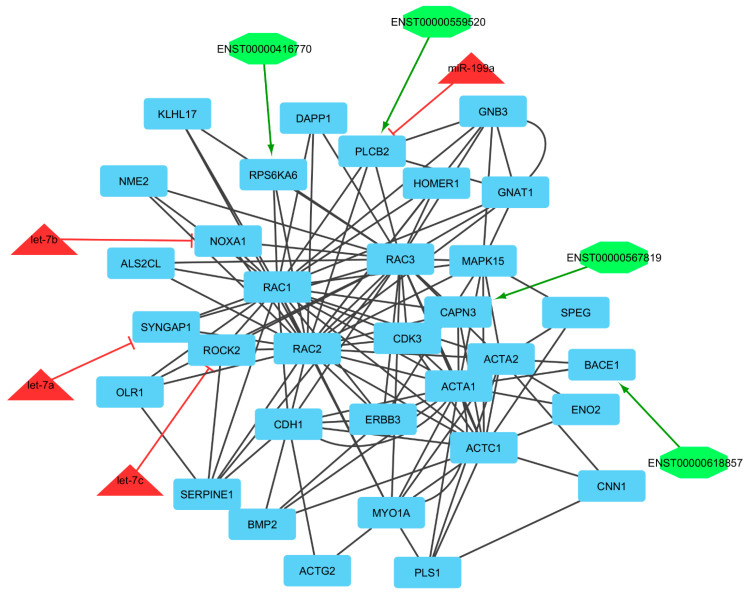
Module 1: 33 mRNAs/genes, 4 lncRNAs, and 4 miRNAs in an interacted network were identified. In this network, the quadrilateral nodes represent mRNAs/genes, octagonal nodes represent lncRNAs, and triangle nodes represent miRNAs. Edges indicate the interactions; black edges represent mRNA–mRNA interactions, green edges represent lncRNA–mRNA interactions, and red edges represent miRNA–mRNA interactions.

**Figure 5 genes-13-01295-f005:**
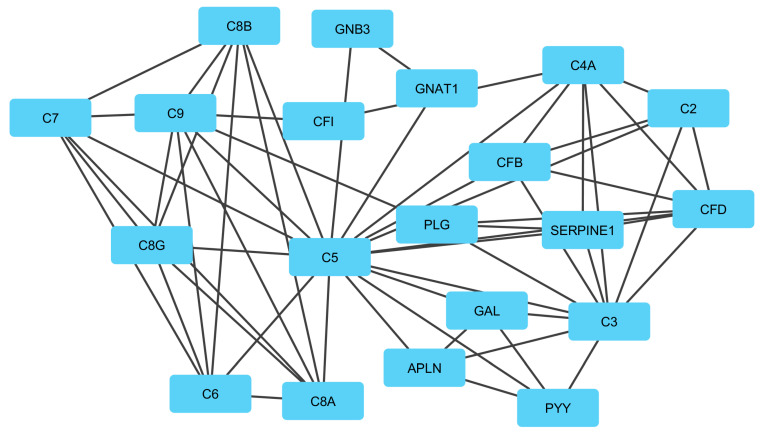
Module 2: 20 genes in an interactive network of genes. In this network, the quadrilateral nodes represent genes, and edges indicate the gene–gene interaction effects.

**Figure 6 genes-13-01295-f006:**
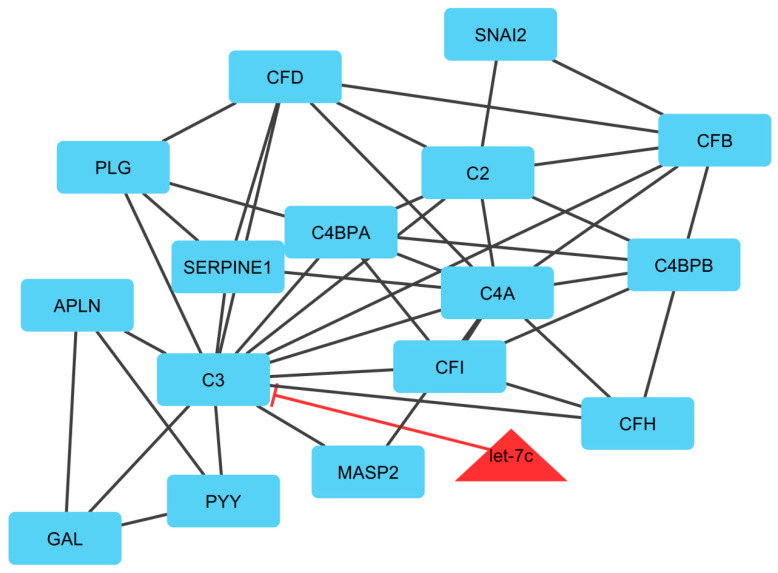
Module 3: 16 mRNAs/genes and 1 miRNA in an interacted network were identified. In this network, the quadrilateral nodes represent mRNAs/genes and triangle nodes represent miRNAs. Edges indicate the interactions; black edges represent mRNA–mRNA interactions and red edges represent miRNA–mRNA interactions.

**Figure 7 genes-13-01295-f007:**
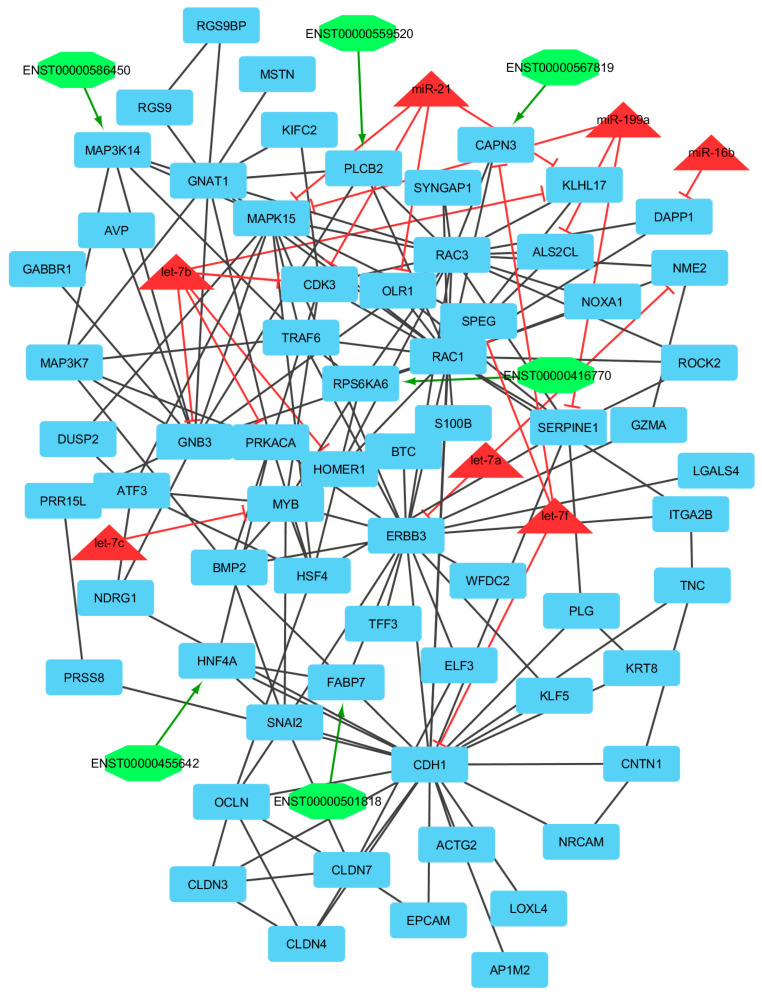
Module 4: 65 mRNAs/genes, 6 lncRNAs, and 7 miRNAs in an interacted network were identified. In this network, the quadrilateral nodes represent mRNAs/genes, octagonal nodes represent lncRNAs, and triangle nodes represent miRNAs. Edges indicate the interactions; black edges represent mRNA–mRNA interactions, green edges represent lncRNA–mRNA interactions, and red edges represent miRNA–mRNA interactions.

**Figure 8 genes-13-01295-f008:**
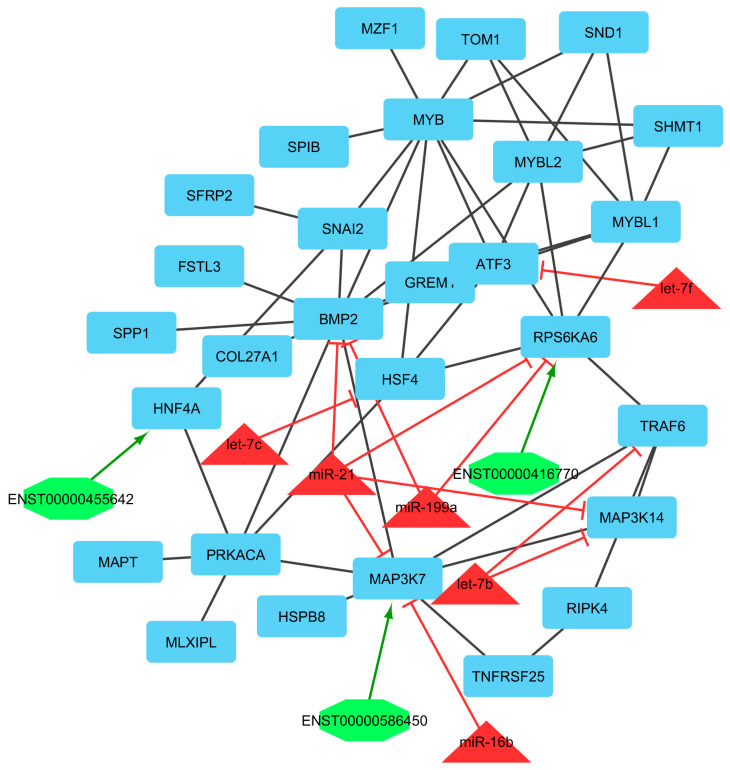
Module 5: 68 mRNAs/genes, 3 lncRNAs, and 6 miRNAs in an interacted network were identified. In this network, the quadrilateral nodes represent mRNAs/genes, octagonal nodes represent lncRNAs, and triangle nodes represent miRNAs. Edges indicate the interactions; black edges represent mRNA–mRNA interactions, green edges represent lncRNA–mRNA interactions, and red edges represent miRNA–mRNA interactions.

**Figure 9 genes-13-01295-f009:**
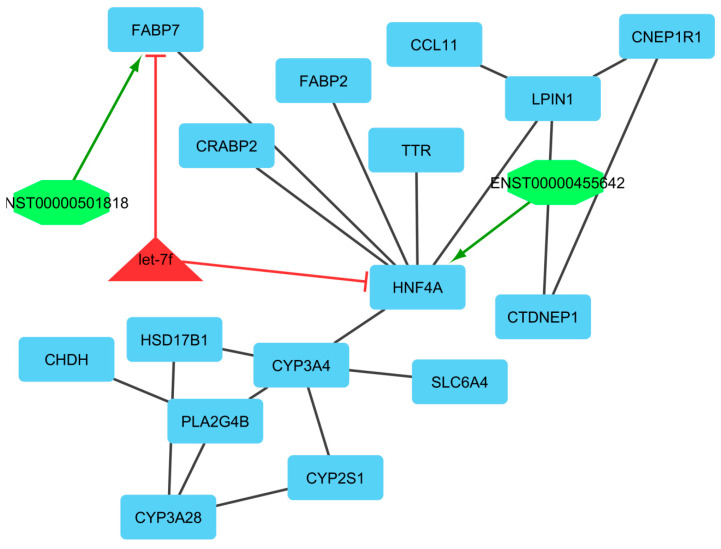
Module 6: 16 mRNAs/genes, 2 lncRNAs, and 1 miRNA in an interacted network were identified. In this network, the quadrilateral nodes represent mRNAs/genes, octagonal nodes represent lncRNAs, and triangle nodes represent miRNAs. Edges indicate the interactions; black edges represent mRNA–mRNA interactions, green edges represent lncRNA–mRNA interactions, and red edges represent miRNA–mRNA interactions.

**Table 1 genes-13-01295-t001:** Basic network statistics of the two generated networks, compared with simulated randomized model networks.

	lncRNA–miRNA–mRNA ceRNA Network	Simulated Barabasi Albert Model (Scale Free)	Simulated Erdos–Renyi Model
Number of nodes	312	312	312
Clustering coefficient	0.108	0.006	0.004
Characteristic path length	4.192	4.947	4.432
Network density	0.006	0.007	0.007
Network centralization	0.070	0.012	0.009

## Data Availability

All sequencing data have been submitted to the National Center for Biotechnology Information (NCBI) Gene Expression Omnibus (GEO). All data can be used without restrictions. Related accession number is GSE159902.

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
