# Peer review of "lncRNA–miRNA–mRNA ceRNA Network Involved in Sheep Prolificacy: An Integrated Approach"

_genes, 2022, doi:10.3390/genes13081295_

Round 1
Reviewer 1 Report
In this manuscript, Sadeghi et al. use NGS technologies to study the lncRNA-miRNA-mRNA ceRNA network between the high- and low-fertility sheeps. The authors identified 264 mRNAs , 14 lncRNA and 34 miRNAs in high-fertility sheeps. The author also discussed the potential functions of these differentially expressed RNAs. However, there are several serious flaws:
1. The authors used the RNeasy kit (Qiagen) to purify/extract the RNAs for sequencing. However, this kit will only recover the RNAs with >200nt length. In this case, how did the authors detect miRNAs in their sequencing data?
2. The authors used |logFC| ≥ 1.5 as the cut-off value to identify the differential expressed genes. However, what are the absolute expression cutoffs for each gene? For example: the authors should also mention the cutoffs like: RPKM >= 0.5 or gene-level read counts >= 10.
3. The first row of Table 1 was miss aligned, it is hard to get the information.
4. Which RNA-RNA database did the author use to predicate the lncRNA-mRNA and miRNA-mRNA interactions? The most of the databases the authors mentioned in the current manuscript was protein-protein interaction.
5. Recently, there are several tools that can be used to analyze RNA-RNA interactions, such as PARIS-seq, RIC-seq, LIGR-seq and so on. The author should use one of these datasets to verify the potential lcRNA-mRNA interactions identified in current study. Or use other experiments (such as: RNA Antisense Purification combined qPCR, RAP–qPCR) to perform verification.
6. The resolution of the figures is very low.
7. The authors should uniform the size of text in Figures. For some of them, it was hard to read.
In summary, it was very interesting to find out the genes and RNAs related to sheep fertility. Without any experimental verification, it is hard to convince readers to believe in the ceRNA network. Could the author verify at least one lncRNA-miRNA-mRNA sub-network?
Author Response
Dear Reviewer
We are grateful for your constructive comments and followed in the revised manuscript. Although, descriptions are mentioned below, but more complete responses are attached in response to the reviewer file.
Point 1: The authors used the RNeasy kit (Qiagen) to purify/extract the RNAs for sequencing. However, this kit will only recover the RNAs with >200nt length. In this case, how did the authors detect miRNAs in their sequencing data?
Response 1: We thank the reviewer for the insightful comments and valuable suggestions. We used miRNeasy procedures and co-purification of total RNA and miRNA, and for simplifying the procedures’ explanation we generally mentioned RNeasy but your recommendation was correct, therefore we have added the detailed explanation of the purification process to the method section.
Point 2: The authors used |logFC| ≥ 1.5 as the cut-off value to identify the differential expressed genes. However, what are the absolute expression cutoffs for each gene? For example: the authors should also mention the cutoffs like: RPKM >= 0.5 or gene-level read counts >= 10.
Response 2: We have applied DESeq2 and a method for differential analysis of count data, using shrinkage estimation for dispersions and fold changes to improve the stability and interpretability of estimates. This enables a more quantitative analysis focused on the strength rather than the mere presence of differential expression. And logFC is calculated based on different factors such as RPKM, therefore almost studies use these cutoffs. For increasing accuracy and identifying key-related hub genes, |logFC| ≥ 1.5 was usually considered (1, 2).
Reference
- Love, M.I., Huber, W. & Anders, S. Moderated estimation of fold change and dispersion for RNA-seq data with DESeq2. Genome Biol 15, 550 (2014). https://doi.org/10.1186/s13059-014-0550-8
- Law CW, Alhamdoosh M, Su S et al. RNA-seq analysis is easy as 1-2-3 with limma, Glimma and edgeR [version 2; peer review: 3 approved]. F1000Research 2016, 5:1408
Point 3: The first row of Table 1 was miss aligned, it is hard to get the information.
Response 3: We have tried to align and centralize the first row of Table 1.
Point 4: Which RNA-RNA database did the author use to predicate the lncRNA-mRNA and miRNA-mRNA interactions? The most of the databases the authors mentioned in the current manuscript was protein-protein interaction.
Response 4: The predicted and validated mRNA genes were identified using miRWalk 3.0 (http://129.206.7.150/) [3,4], a comprehensive atlas of microRNA-target interactions tool which integrates 12 miRNA-target prediction tools. And lncRNA-mRNA interactions were predicted by the NONCODE database [5] (http://www.noncode.org/) and the LNCipedia database [6] (https://lncipedia.org).
Reference
- Dweep H, Gretz N, Sticht C. miRWalk database for miRNA-target interactions. Methods in molecular biology. 2014;1182:289–305. pmid:25055920.
- Sticht C, De La Torre C, Parveen A, Gretz N (2018) miRWalk: An online resource for prediction of microRNA binding sites. PLoS ONE 13(10): e0206239.
- Zhao, Y.; Li, H.; Fang, S.; Kang, Y.; Wu, W.; Hao, Y.; et al. Noncode 2016: an informative and valuable data source of long non-coding rnas. Nucleic Acids Res. 2015, 44, D203–D208.
- Volders, P. J.; Anckaert, J.; Verheggen, K.; Nuytens, J.; Martens, L.; Mestdagh, P.; et al. Lncipedia 5: towards a reference set of human long non-coding rnas. Nucleic Acids Res. 2018, 47, D135–D139.
Point 5: Recently, there are several tools that can be used to analyze RNA-RNA interactions, such as PARIS-seq, RIC-seq, LIGR-seq and so on. The author should use one of these datasets to verify the potential lcRNA-mRNA interactions identified in current study. Or use other experiments (such as: RNA Antisense Purification combined qPCR, RAP–qPCR) to perform verification.
Response 5: We have utilized miRWalk 3.0 database that presents predicted and validated information on miRNA-target interaction. We have tried to use just validated interactions (experimentally). Such a resource enables researchers to validate new targets of miRNA not only on 3′-UTR, but also on the other regions of all known genes. This database integrates 12 interaction tools and databases for example; TargetScan, miRBase, miRDB, miRTarBase, TarPmiR, etc. for other RNA-RNA interactions, we have used the most comprehensive lncRNA interactions tools and databases mentioned above. As well, for in situ validation of DEG, IGV was used. Although, it can also help us to better appreciate the molecular mechanisms responsible for the generation of aberrant transcripts that are common in malignant states. Tremendous progress has been done in developing multiple methods to map RNA interactions at different levels. However, the abundance of data obtained with these technologies also naturally brings with it challenges in interpreting this information. Thus, comprehensive elucidations of the properties of RNA interactions are critical for our appreciation of the complexities of the function and regulation of RNA and protein-coding mRNAs. Finally, we have tried to add qRT-PCR analysis for two mRNAs that were up-regulated and two mRNA that were down-regulated to increase the reproducibility and reliability of RNA-Seq results.
Point 6: The resolution of the figures is very low.
Response 6: We have tried to increase figures resolution.
Point 7: The authors should uniform the size of text in Figures. For some of them, it was hard to read.
Response 7: We have tried to increase figures resolution and uniform them but when word file is converting to PDF file, changed most of them.
Point 8: In summary, it was very interesting to find out the genes and RNAs related to sheep fertility. Without any experimental verification, it is hard to convince readers to believe in the ceRNA network. Could the author verify at least one lncRNA-miRNA-mRNA sub-network?
Response 8: We are grateful for your constructive comments and followed in the revised manuscript. We have tried to apply multi-partite approaches for this study and this study is the first study in its aspect. As well, we have tried to simplify the workflow of the study. As you know bioinformatics and systems biology studies, is use different RNAs and integrate them for surveying comprehensive interpretations, so we have used sequencing data and pre and post-processing on these data. So, as you know understanding interaction can help to a better understanding of post transcription changes and identified RNAs can be used for further studies such as breeding strategies, genome editing and etc. The previous studies just have surveyed and attended level of genotype in the event that we have considered gene co-expression, PPI, literature mining, pathway interactions, and all interaction between different RNA databases. However, we agree with you and believe further research is needed and in this research, we have tried to apply differently in situ validation tools for verifying lncRNA-miRNA-mRNA networks.

Reviewer 2 Report
Comment 1:
------------
Romanov and Baluchi sheep are two different breeds? Or are they derived from a common ancestor? It is well known that samples from sheep with different genetic backgrounds can seriously interfere with the accuracy of the data. According to the results of this study, there were 264 differentially expressed genes, but no known genes related to sheep reproductive traits were identified. Such as CCNB2 and SLC8A3 (Oocyte development); PRLR (reproductive performance); GDF9, BMP15 and BMPR1B ( Ovulation rate & sterility). Therefore, the reliability of the data is questionable.
Comment 2:
------------
The CeRNA is a gene expression regulation mode, in which transcripts sharing miRNA binding sites compete to bind the same miRNA, thereby regulating each other's expression. In this manuscript, ceRNA was repeatedly mentioned, but the binding interaction between lncRNA and miRNA was not shown in 6 key modules (figure4-9). For example, which lncRNAs competitively bind miR-199a to regulate PLCB2 expression? In addition, the mechanism of ceRNA that may affect the reproductive traits of sheep was not mentioned in the discussion
Comment 3:
------------
Authors should perform further validation or follow-on biological studies to support the conclusions of the study. For example, qPCR was used to verify the differential expression of lncRNA, miRNA and mRNA. Luciferase reporting system was used to verify the targeting relationship between miRNA and mRNA etc.
Comment 4:
------------
An improved description of the corpus luteum tissue sample is required. Does the sample include the granulosa lutein cell or theca lutein cell? Sample location, size, etc.
Comment 5:
------------
The clarity of figures 2 and 3 needs to be improved. Moreover, Supplementary Table 1 was not mentioned in the whole manuscript
Comment 6:
------------
Line 246,Remove the first line indent by 2 characters
Line 424,“lnvRNA” should be replaced with “lncRNA”
Author Response
Dear Reviewer
We are grateful for your constructive comments and followed in the revised manuscript. Although, descriptions are mentioned below, but more complete responses are attached in response to the reviewer file.
Point 1: Romanov and Baluchi sheep are two different breeds? Or are they derived from a common ancestor? It is well known that samples from sheep with different genetic backgrounds can seriously interfere with the accuracy of the data. According to the results of this study, there were 264 differentially expressed genes, but no known genes related to sheep reproductive traits were identified. Such as CCNB2 and SLC8A3 (Oocyte development); PRLR (reproductive performance); GDF9, BMP15 and BMPR1B (Ovulation rate & sterility). Therefore, the reliability of the data is questionable.
Response 1: We are grateful for the suggestion. Indeed, the use of the same breed maybe leads to more accurate results. However, these two sheep breeds are different, but in systemic studies, thresholds are usually considered that have the most genetic and phenotypic differences. Even in many studies, the expression difference between the two different species is done for identifying genes that in any experimental condition showed a change in a special biological process.
In this study, we considered two breeds of a species whose genetic background is very common, and considering that the biological process is a molecular cellular process, it is easy to extract a stable (static) model for it using integrated approaches. Because cellular processes, especially at the molecular level, are similar in up to 99% of cases, even among different animal species, and it cannot be said that due to changes in environmental conditions or even species, no argument can be made for how the process occurs. As well, the related genes may be identified in different data sets that depend on the target tissues and the main purpose of this study is the corpus luteum expression profile. In some of the mentioned genes, their family genes were identified, for example, for SLC8A3 (Oocyte development) gene, SLC12A3, SLC44A4, SLC9A5, SLC13A3 were identified and also about other genes. Thus, complex biological relationships may involve genes in a gene cluster and be controlled by intergenic interactions. However, most of these genes have been reported in the list of DEG and have not been included in the final list after filtration, especially logFC.
Point 2: The CeRNA is a gene expression regulation mode, in which transcripts sharing miRNA binding sites compete to bind the same miRNA, thereby regulating each other's expression. In this manuscript, ceRNA was repeatedly mentioned, but the binding interaction between lncRNA and miRNA was not shown in 6 key modules (figure4-9). For example, which lncRNAs competitively bind miR-199a to regulate PLCB2 expression? In addition, the mechanism of ceRNA that may affect the reproductive traits of sheep was not mentioned in the discussion
Response 2: We have tried to discuss and add more explanations about the competitive bind between lncRNA and miRNA. Although, restricted databases and tools are available even for Homo sapiens, and existed tools consider miRNA-gene or lncRNA-gene interactions. But we have obtained outstanding results in terms of levels of expression and results showed there was a negative correlation between up or down expression between mRNA, miRNA, and lncRNA and approximately verifying achievements.
Point 3: Authors should perform further validation or follow-on biological studies to support the conclusions of the study. For example, qPCR was used to verify the differential expression of lncRNA, miRNA and mRNA. Luciferase reporting system was used to verify the targeting relationship between miRNA and mRNA etc.
Response 3: We are grateful for the reviewer’s constructive comments. We have tried to improve the revised manuscript. In RNA-Seq technology, we have read and the measuring changes in the expression level of genes (fold change) and even the sequence of samples. And with the use of different software and websites (such as IGV), we can use these for validation, however, your recommendation is so valuable and increases results reliability (1-5). And the other hand, one of the main reasons was restricted financial support. In this regard, we have tried to add qRT-PCR analysis for two mRNAs that were up-regulated and two mRNAs that were down-regulated to increase the reproducibility and reliability of RNA-Seq results. In addition, in the Materials and Methods section, we have added a sub-section for validation.
References:
- Hasankhani, A., Bahrami, A., Sheybani, N., Fatehi, F., Abadeh, R., Ghaem Maghami Farahani, H., Bahreini Behzadi, M.R., Javanmard, G., Isapour, S., Khadem, H. and Barkema, H.W., 2021. Integrated Network Analysis to Identify Key Modules and Potential Hub Genes Involved in Bovine Respiratory Disease: A Systems Biology Approach. Frontiers in Genetics, 2001.
- Naserkheil, M., Ghafouri, F., Zakizadeh, S., Pirany, N., Manzari, Z., Ghorbani, S., Banabazi, M.H., Bakhtiarizadeh, M.R., Huq, M., Park, M.N. and Barkema, H.W., 2022. Multi-Omics Integration and Network Analysis Reveal Potential Hub Genes and Genetic Mechanisms Regulating Bovine Mastitis. Current Issues in Molecular Biology, 44(1), 309-328.
- Ghafouri, F., Bahrami, A., Sadeghi, M., Miraei-Ashtiani, S.R., Bakherad, M., Barkema, H.W. and Larose, S., 2021. Omics multi-layers networks provide novel mechanistic and functional insights into fat storage and lipid metabolism in poultry. Frontiers in Genetics, 12.
- Wang, Y., Niu, Z., Zeng, Z., Jiang, Y., Jiang, Y., Ding, Y., Tang, S., Shi, H. and Ding, X., 2020. Using High-Density SNP Array to Reveal Selection Signatures Related to Prolificacy in Chinese and Kazakhstan Sheep Breeds. Animals, 10(9), 1633.
- Bahrami, A., Miraie-Ashtiani, S.R., Sadeghi, M. and Najafi, A., 2017. miRNA-mRNA network involved in folliculogenesis interactome: systems biology approach. Reproduction, 154(1), 51-65.
Point 4: An improved description of the corpus luteum tissue sample is required. Does the sample include the granulosa lutein cell or theca lutein cell? Sample location, size, etc.
Response 4: We have tried to add details of sampling to the method section. In the biopsy sample, the mean luteal tissue area was 425 mm2 (range, 313 to 524 mm2). Some corpora lutea contained a central luteal cavity with an area of 335 mm2. A total of 59 biopsy attempts were made; 54 (92%) resulted in obtaining a luteal tissue specimen at the first attempt. The average size of the obtained biopsy core was 1 mm in diameter and 10.8 mm in length (range, 10 to 12 mm), and the average weight was 4.4 mg (range, 1.8 to 7.5 mg).
Point 5: The clarity of figures 2 and 3 needs to be improved. Moreover, Supplementary Table 1 was not mentioned in the whole manuscript
Response 5: We have tried to increase figures resolution and uniform them but when word file is converting to PDF file, changed most of them. As well, we have mentioned Supplementary Table 1 in the context.
Point 6: Line 246,Remove the first line indent by 2 characters
Line 424,“lnvRNA” should be replaced with “lncRNA”
Response 6: We have removed the first line between two characters.
And, we have replaced “lnvRNA” with “lncRNA”

Reviewer 3 Report
In this manuscript, whole-transcriptome analysis of the corpus luteum samples from high-fertility and low-fertility sheep were performed, mRNAs/genes, lncRNAs, miRNAs were identified into 6 modules, these RNAs involved in different biological processes but no
validated processes were associated with prolificacy development. However, the ceRNA network may give a chance to study genetic traits such as reproductive process.
1. Animal background was not described clearly enough in the manuscript. The sampling season and age were not mentioned in Methods section. What are the details of phenotypic recording, such as fetal times, litter size, How do authors consider these basis.
2 .Two sheep breeds with differentially genetic background and phenotype(high fecundity and low fecundity)were selected for transcriptome analysis, however, animals with same genetic background and different phenotype may be more ideal materials for this study, for example, two groups of sheep from the same breeds, sheep from one group have high litter size while the other have low litter size. Does the author have any views of this question?
3. Resolutions of all figures is not high enough, the Fig1 is a screen capture, the author should supply formal figures according to submission request.
4. This manuscript aimed to understand the molecular mechanisms responsible for fertility. However, the results showed no major genes related to reproductive traits, such as B4GALNT2, BMPR1B, BMP15, and GDF9, were detected in ceRNA network. Does the author have any views of the results?
5. In Methods, tissue biopsy needles were used for luteal biopsy, it is not clear that how to select the corpus luteum, the standard of selection? How about the diameter of the needle and how deep the needle stabbed into the luteum.
Author Response
Dear Reviewer
We are grateful for your constructive comments and followed in the revised manuscript. Although, descriptions are mentioned below, but more complete responses are attached in response to the reviewer file.
In this manuscript, whole-transcriptome analysis of the corpus luteum samples from high-fertility and low-fertility sheep were performed, mRNAs/genes, lncRNAs, miRNAs were identified into 6 modules, these RNAs involved in different biological processes but no validated processes were associated with prolificacy development. However, the ceRNA network may give a chance to study genetic traits such as reproductive process.
In RNA-Seq technology, we have read and the measuring change in the expression level of genes (fold change) and even the sequence of samples. And with the use of different software and websites (such as IGV), we can use these for validation, however, your recommendation is so valuable and increases results reliability (1-5). And the other hand, one of the main reasons was restricted financial support. In this regard, we have tried to add qRT-PCR analysis for two mRNAs that were up-regulated and two mRNAs that were down-regulated to increase the reproducibility and reliability of RNA-Seq results. In addition, in the Materials and Methods section, we have added a sub-section for validation.
References:
- Hasankhani, A., Bahrami, A., Sheybani, N., Fatehi, F., Abadeh, R., Ghaem Maghami Farahani, H., Bahreini Behzadi, M.R., Javanmard, G., Isapour, S., Khadem, H. and Barkema, H.W., 2021. Integrated Network Analysis to Identify Key Modules and Potential Hub Genes Involved in Bovine Respiratory Disease: A Systems Biology Approach. Frontiers in Genetics, 2001.
- Naserkheil, M., Ghafouri, F., Zakizadeh, S., Pirany, N., Manzari, Z., Ghorbani, S., Banabazi, M.H., Bakhtiarizadeh, M.R., Huq, M., Park, M.N. and Barkema, H.W., 2022. Multi-Omics Integration and Network Analysis Reveal Potential Hub Genes and Genetic Mechanisms Regulating Bovine Mastitis. Current Issues in Molecular Biology, 44(1), 309-328.
- Ghafouri, F., Bahrami, A., Sadeghi, M., Miraei-Ashtiani, S.R., Bakherad, M., Barkema, H.W. and Larose, S., 2021. Omics multi-layers networks provide novel mechanistic and functional insights into fat storage and lipid metabolism in poultry. Frontiers in Genetics, 12.
- Wang, Y., Niu, Z., Zeng, Z., Jiang, Y., Jiang, Y., Ding, Y., Tang, S., Shi, H. and Ding, X., 2020. Using High-Density SNP Array to Reveal Selection Signatures Related to Prolificacy in Chinese and Kazakhstan Sheep Breeds. Animals, 10(9), 1633.
- Bahrami, A., Miraie-Ashtiani, S.R., Sadeghi, M. and Najafi, A., 2017. miRNA-mRNA network involved in folliculogenesis interactome: systems biology approach. Reproduction, 154(1), 51-65.
Point 1: Animal background was not described clearly enough in the manuscript. The sampling season and age were not mentioned in Methods section. What are the details of phenotypic recording, such as fetal times, litter size, How do authors consider these basis.
Response 1: We have tried to add data for the phenotype of litter size of ewes as Supplementary Table 1.
Point 2: Two sheep breeds with differentially genetic background and phenotype(high fecundity and low fecundity)were selected for transcriptome analysis, however, animals with same genetic background and different phenotype may be more ideal materials for this study, for example, two groups of sheep from the same breeds, sheep from one group have high litter size while the other have low litter size. Does the author have any views of this question?
Response 2: We are grateful for the suggestion. Indeed, the use of the same breed maybe leads to more accurate results. However, these two sheep breeds are different, but in systemic studies, thresholds are usually considered that have the most genetic and phenotypic differences. Even in many studies, the expression difference between the two different species is done for identifying genes that in any experimental condition showed a change in a special biological process. In this study, we considered two breeds of a species whose genetic background is very common, and considering that the biological process is a molecular cellular process, it is easy to extract a stable (static) model for it using integrated approaches. Because cellular processes, especially at the molecular level, are similar in up to 99% of cases, even among different animal species, and it cannot be said that due to changes in environmental conditions or even species, no argument can be made for how the process occurs. Finally, most studies consider a wide range of different breeds to identify genetic and transcriptomic differences (6-13).
Reference
- Davis, G. H., Balakrishnan, L., Ross, I. K., Wilson, T., Galloway, S. M., Lumsden, B. M., et al. (2006a). Investigation of the Booroola (FecB) and Inverdale (FecXI) mutations in 21 prolific breeds and strains of sheep sampled in 13 countries. Anim. Reprod. Sci. 92, 87–96. doi: 10.1016/j.anireprosci.2005.06.001
- Peng, W. F., Xu, S. S., Ren, X., Lv, F. H., Xie, X. L., Zhao, Y. X., et al. (2017). A genome-wide association study reveals candidate genes for the supernumerary nipple phenotype in sheep (Ovis aries). Anim. Genet. 48, 570–579. doi: 10.1111/age.12575
- Miao, X., Luo, Q., Zhao, H., and Qin, X. (2016). Ovarian transcriptomic study reveals the differential regulation of miRNAs and lncRNAs related to fecundity in different sheep. Sci. Rep. 6:35299. doi: 10.1038/srep 35299
- Zhang, C. S., Geng, L. Y., Du, L. X., Liu, Z. Z., Fu, Z. X., Feng, M. S., et al. (2011). Polymorphic study of FecX(G), FecG(H) and Fec(B) mutations in four domestic sheep breeds in the Lower Yellow River Valley of China. J. Anim. Vet. Adv. 10, 2198–2201. doi: 10.3923/javaa.2011.2198.2201
- Xu S-S, Gao L, Xie X-L, Ren Y-L, Shen Z-Q, Wang F, Shen M, Eyþórsdóttir E, Hallsson JH, Kiseleva T, Kantanen J and Li M-H (2018) Genome-Wide Association Analyses Highlight the Potential for Different Genetic Mechanisms for Litter Size Among Sheep Breeds. Front. Genet. 9:118.
- Wang, J.; Ren, Q.; Hua, L.; Chen, J.; Zhang, J.; Bai, H.; Li, H.; Xu, B.; Shi, Z.; Cao, H.; et al. Comprehensive Analysis of Differentially Expressed mRNA, lncRNA and circRNA and Their ceRNA Networks in the Longissimus Dorsi Muscle of Two Different Pig Breeds. Int. J. Mol. Sci. 2019, 20, 1107.
- Shi, T.; Hu,W.; Hou, H.; Zhao, Z.; Shang, M.; Zhang, L. Identification and Comparative Analysis of Long Non-Coding RNA in the Skeletal Muscle of Two Dezhou Donkey Strains. Genes 2020, 11, 508.
- Sun, J.; Xie, M.; Huang, Z.; Li, H.; Chen, T.; Sun, R.; Wang, J.; Xi, Q.; Wu, T.; Zhang, Y. Integrated analysis of non-coding RNA and mRNA expression profiles of 2 pig breeds differing in muscle traits1,2. J. Anim. Sci. 2017, 95, 1092–1103.
Point 3: Resolutions of all figures is not high enough, the Fig1 is a screen capture, the author should supply formal figures according to submission request.
Response 3: We have tried to increase figures resolution and uniform them but when word file is converting to PDF file, changed most of them.
Point 4: This manuscript aimed to understand the molecular mechanisms responsible for fertility. However, the results showed no major genes related to reproductive traits, such as B4GALNT2, BMPR1B, BMP15, and GDF9, were detected in ceRNA network. Does the author have any views of the results?
Response 4: The related genes may be identified in different data sets depending on the target tissues and the main purpose of this study is the corpus luteum expression profile. In some of the mentioned genes, their family genes were identified, for example, for SLC8A3 (Oocyte development) gene, SLC12A3, SLC44A4, SLC9A5, SLC13A3 were identified, or BMP2 and BMP7 belong to BMP family genes (Ovulation rate & sterility) and similar to BMP15, BMPR1B genes function. Thus, complex biological relationships may involve genes in a gene cluster and be controlled by intergenic interactions. However, most of these genes have been reported in the list of DEG and have not been included in the final list after filtration, especially logFC thresholds.
Point 5: In Methods, tissue biopsy needles were used for luteal biopsy, it is not clear that how to select the corpus luteum, the standard of selection? How about the diameter of the needle and how deep the needle stabbed into the luteum.
Response 5: We have tried to add details of sampling to the method section. Although we have mentioned that Kot et al. (14,15) described biopsy procedures in detail:
The luteal biopsy was performed using a tissue biopsy needle (Ovum Pick-up instrument that has been equipped with 48 cm long, trocar tip; SABD-1648-15-T; US BiopsySABD-1648-15-T). In the biopsy sample, the mean luteal tissue area was 425 mm2 (range, 313 to 524 mm2). Some corpora lutea contained a central luteal cavity with an area of 335 mm2. A total of 59 biopsy attempts were made; 54 (92%) resulted in obtaining a luteal tissue specimen at the first attempt. The average size of the obtained biopsy core was 1 mm in diameter and 10.8 mm in length (range, 10 to 12 mm), and the average weight was 4.4 mg (range, 1.8 to 7.5 mg).
Reference
- Kot K, Anderson LE, Tsai SJ, Wiltbank MC, Ginther OJ. Transvaginal, ultrasound-guided biopsy of the corpus luteum in cattle. Theriogenology 1999; 52(6):987–993.
- Henry, L., Fransolet, M., Labied, S. et al. Supplementation of transport and freezing media with anti-apoptotic drugs improves ovarian cortex survival. J Ovarian Res 9, 4 (2016). https://doi.org/10.1186/s13048-016-0216-0

Round 2
Reviewer 1 Report
The authors still didn't explain well how they can extract miRNA using RNeasy Kit. Also, it would be great if the authors could add the base-pair interaction patterns of miRNA-target RNA to related figures.
Author Response
Point 1: The authors still didn't explain well how they can extract miRNA using RNeasy Kit. Also, it would be great if the authors could add the base-pair interaction patterns of miRNA-target RNA to related figures.
Response 1: We thank the reviewer for the insightful comments and valuable suggestions. Further to previous explaination, we have corrected miss-typing and RNeasy replaced by miRNeasy kit in the text and explained that “we used miRNeasy procedures and co-purification of total RNA and miRNA based on below figure, and for simplifying the procedures’ explanation we generally mentioned RNeasy but your recommendation was correct, therefore we have added the detailed explanation of the purification process to the method section”. As well, we have tried to show miRNA-target RNA interaction by changing edges lables. And generally, each miRNA regulates a specific set of mRNA “targets” by binding to complementary sequences in their 3′ untranslated region (1).
Reference
- Carmel I, Shomron N, Heifetz Y. Does base-pairing strength play a role in microRNA repression? RNA. 2012 Nov;18(11):1947-56. doi: 10.1261/rna.032185.111.

Reviewer 2 Report
Comment 1:
------------
I still think ceRNA is the focus and highlight of this manuscript, not just miRNA-gene or lncRNA-gene interactions. The ceRNA hypothesis is mentioned in lines 439-441 of the manuscript. The lncRNA-miRNA-mRNA ceRNA network was constructed in the material and method (Line 242). However, no results showed that lncRNAs competitively bind miRNAs in this manuscript. Therefore, lncRNA-miRNA-gene interactions should be displayed in the results. Please refer to the figure below.
Figure: The lncRNA-miRNA-mRNA ceRNA regulatory network. Rectangles represent differentially expressed miRNAs, ellipses represent differentially expressed mRNAs, and the triangle represent differentially expressed lncRNAs. The gray lines represent the interactions between diverse RNAs. Note: The source of the “Figure” is “figure 7” of references [73] of this manuscript.
Comment 2:
------------
All supplementary tables (Supplementary Table 1-8) in the manuscript have no headings or titles, notes, etc. For example, in supplementary table 7 without any title and comments, I can't understand what the author is trying to show. This greatly affects the quality of the manuscript.
Comment 3:
------------
Statistical data in supplementary table1 have no labeling error value. Such as 2.3±?(Avg. Lambs per year in Rm)
Comment 4:
------------
In supplementary table 4, 3 lncRNAs with fold change > 1.5 and 11 lncRNAs with fold change < -1.5 were shown, which was inconsistent with the description of lines 299-301(8 and 6 lncRNAs) in the manuscript.
Comment 5:
------------
Supplementary table 5 should be simplified to retain useful information. fThe fold change value and P value are not found in supplementary table 5.
Comment 6:
------------
ACTB should not be abbreviated when it first appears.
Author Response
Point 1: I still think ceRNA is the focus and highlight of this manuscript, not just miRNA-gene or lncRNA-gene interactions. The ceRNA hypothesis is mentioned in lines 439-441 of the manuscript. The lncRNA-miRNA-mRNA ceRNA network was constructed in the material and method (Line 242). However, no results showed that lncRNAs competitively bind miRNAs in this manuscript. Therefore, lncRNA-miRNA-gene interactions should be displayed in the results. Please refer to the figure below.
Response 1: We are grateful for the suggestion. Indeed, the use of the same breed maybe leads to more accurate results. Despite we could not find any figure and reference 73 in the comment but we have tried to discuss and add more explanations about the competitive bind between lncRNA and miRNA. Although, restricted databases and tools are available even for Homo sapiens, and existed tools consider miRNA-gene or lncRNA-gene interactions. But we have obtained outstanding results in terms of levels of expression and results showed there was a negative correlation between up or down expression between mRNA, miRNA, and lncRNA and approximately verifying achievements. As well, we have tried to show miRNA-target RNA interaction by changing edges lables. And generally, each miRNA regulates a specific set of mRNA “targets” by binding to complementary sequences in their 3′ untranslated region (1).
Reference
- Carmel I, Shomron N, Heifetz Y. Does base-pairing strength play a role in microRNA repression? RNA. 2012 Nov;18(11):1947-56. doi: 10.1261/rna.032185.111.
Point 2: All supplementary tables (Supplementary Table 1-8) in the manuscript have no headings or titles, notes, etc. For example, in supplementary table 7 without any title and comments, I can't understand what the author is trying to show. This greatly affects the quality of the manuscript.
Response 2: We have tried to add related titles to all supplementary tables.
Point 3: Statistical data in supplementary table1 have no labeling error value. Such as 2.3±?(Avg. Lambs per year in Rm)
Response 3: We have tried to add labeling error value.
Point 4: In supplementary table 4, 3 lncRNAs with fold change > 1.5 and 11 lncRNAs with fold change < -1.5 were shown, which was inconsistent with the description of lines 299-301(8 and 6 lncRNAs) in the manuscript.
Response 4: We have tried to correct occurred mistyping in the result section and 3 lncRNAs with fold change > 1.5 and 11 lncRNAs with fold change < -1.5 are correct.
Point 5: Supplementary table 5 should be simplified to retain useful information. fThe fold change value and P value are not found in supplementary table 5.
Response 5: We have added the fold change and P value to supplementary table 5.
Point 6: ACTB should not be abbreviated when it first appears.
Response 6: We have added related abbreviation in the method section

Reviewer 3 Report
The figure 2 could be deleted and added into supplymentary files.
Author Response
Point 1: The figure 2 could be deleted and added into supplementary files.
Response 1: We have added Figure 2 into supplementary files.
